# Regulation of CRE-Dependent Transcriptional Activity in a Mouse Suprachiasmatic Nucleus Cell Line

**DOI:** 10.3390/ijms232012226

**Published:** 2022-10-13

**Authors:** Monica Langiu, Philipp Bechstein, Sonja Neumann, Gabriele Spohn, Erik Maronde

**Affiliations:** 1Institut für Anatomie II, Dr. Senckenbergische Anatomie, Goethe-Universität Frankfurt, Theodor-Stern-Kai-7, 60590 Frankfurt, Germany; 2Monash Institute of Pharmaceutical Science, Monash University, 399 Royal Parade, Parkville, VIC 3052, Australia; 3German Red Cross Blood Service, Sandhofstrasse-1, 60528 Frankfurt, Germany

**Keywords:** CRE, SCN, pCREB, signalling pathways, PKA, PKC

## Abstract

We evaluated the signalling framework of immortalized cells from the hypothalamic suprachiasmatic nucleus (SCN) of the mouse. We selected a vasoactive intestinal peptide (VIP)-positive sub-clone of immortalized mouse SCN-cells stably expressing a cAMP-regulated-element (CRE)-luciferase construct named SCNCRE. We characterized these cells in terms of their status as neuronal cells, as well as for important components of the cAMP-dependent signal transduction pathway and compared them to SCN ex vivo. SCNCRE cells were treated with agents that modulate different intracellular signalling pathways to investigate their potency and timing for transcriptional CRE-dependent signalling. Several activating pathways modulate SCN neuronal signalling via the cAMP-regulated-element (CRE: TGACGCTA) and phosphorylation of transcription factors such as cAMP-regulated-element-binding protein (CREB). CRE-luciferase activity induced by different cAMP-signalling pathway-modulating agents displayed a variety of substance-specific dose and time-dependent profiles and interactions relevant to the regulation of SCN physiology. Moreover, the induction of the protein kinase C (PKC) pathway by phorbol ester application modulates the CRE-dependent signalling pathway as well. In conclusion, the cAMP/PKA- and the PKC-regulated pathways individually and in combination modulate the final CRE-dependent transcriptional output.

## 1. Introduction

Transcriptional activation in response to an elevation of intracellular adenosine-3′,5′-(cyclic)monophosphate (cAMP) is a widespread phenomenon depending, among other factors, on the dissociation of PKA into regulatory and catalytic subunits and transport of the catalytic subunit of PKA from the cytosol to the nucleus [1,2]. In the suprachiasmatic nucleus (SCN) of the hypothalamus cAMP/PKA-dependent phosphorylation of cAMP-regulated-element-binding protein (CREB) has been shown to be involved in the amplitude and phase shift of circadian rhythms [3]. The SCN was one of the first neuronal systems in which CREB and CRE (TGACGCTA) regulated transcriptional activity have been shown to play a physiological role, namely clock genes induction and phase shift of their circadian expression [3]. 

In the SCN, as well as in other cells and tissues, the second messenger cAMP is also regarded as critical for the transcriptional activation of many genes containing a CRE or similar regulatory elements [4,5,6].

In the SCN, in vivo cAMP levels vary with a diurnal (light-dark dependent) and circadian (“free running”) rhythm that has a phase length of approximately 24 h. However, not all neurons inside the SCN display circadian rhythms. In the non-circadian cells of the SCN, cAMP may be regulated by neurotransmitters and hormones, and these cells may contribute to e.g., sensing of the light-regulated variation of neurotransmitters by shifting of the phase of the circadian clock [7,8].

The SCN, as the site of the central circadian clock of mammals, consists, beside glial cells, of parvocellular neuronal cell populations, which are characterized by their location within the SCN and the neurotransmitter(s) they express and/or secrete [9]. One such subpopulation of SCN neurons expresses and produces the neuropeptide vasoactive intestinal polypeptide (VIP) [10,11,12]. 

We characterized the fine-tuning of cAMP signalling in an immortalized mouse SCN cell line [13], expressing VIP, which was stably transfected with a CRE luciferase reporter plasmid. 

We used this cell line named SCNCRE for the investigation of the dose- and time-dependency of the response toward different agents relevant to the physiology of this particular VIP-positive SCN neuronal cell population.

Among the parameters tested for the different cAMP-elevating agents known or postulated to modulate SCN function were the dose dependency, the timing of the onset (first significant elevation over untreated or vehicle controls), and the maximum or peak of the reaction. In the course of these experiments, participation of adenylyl cyclase (the enzyme catalysing the synthesis of cAMP from ATP), cAMP-dependent protein kinase (PKA) [14,15] and other cAMP effector proteins, such as cyclic 3′,5′-phosphodiesterases (PDE) [16] and organic anion exchangers (OAE) were investigated in detail. 

Other potential targets for cAMP, like cyclic nucleotide-gated ion channels (CNG; [9]) and exchange factors directly activated by cAMP (EPAC; [17,18]), were also evaluated. Finally, we investigated interactions among the different cAMP-modulating components and cross-talk with the protein kinase C (PKC) pathway on the level of CRE-dependent transcriptional activity.

## 2. Results

### 2.1. Characterization of SCNCRE Cells

SCNCRE cells were derived from an immortalized mouse hypothalamic suprachiasmatic nucleus cell line [13]. Using an antibody against “vasoactive intestinal peptide” (VIP) in immunocytochemical (ICC) staining, most SCNCRE cells displayed VIP-positive granular material in the cytoplasm (Figure 1A,B). These cells were strongly immunopositive for the neuropeptide VIP but only very weakly positive for arginine vasopressin (AVP) (as shown in Figure 1A,B), suggesting that SCNCRE cells represent a subpopulation of VIP-expressing parvocellular neuronal cells from the SCN. This is underscored by our finding that SCNCRE displays a strong signal for the native, unprocessed form of the vasoactive intestinal peptide, proVIP (17 kDa), in the cytoplasm as shown in Western blot experiments (Figure 1C, lanes 5/6; see also Appendix A). 

The state of the SCNCRE cells in terms of differentiation as neuronal cells was established by the findings that they stain positive for βIII-Tubulin similar to extracts of the native mouse SCN ex vivo (Appendix A) and MAP2B (Appendix A). Neither SCN ex vivo nor SCNCRE cells show signals for Nestin or Doublecortin, marker proteins for undifferentiated or maturing neuronal cells (Appendix A). The astrocyte marker glial fibrillary acidic protein (GFAP) was, as expected, present in the SCN ex vivo extracts, but not in SCNCRE cell extracts (Appendix A).

### 2.2. Characteristics of PKA and Protein Phosphorylation in SCN Ex Vivo and SCNCRE Cells

SCNCRE cells and SCN ex vivo displayed robust signals for both PKA regulatory subunits RIIα and RIIβ (Figure 1C lanes 9–12 and Figure 1D lanes 1–4). However, we were not able to detect type RIα or RIβ in both SCNCRE cells or SCN extracts (as exemplarily shown in Figure 1C, lanes 1 to 4). 

SCNCRE cells treated with forskolin displayed increased levels of phosphorylated proteins as can be visualized by an antiserum against the protein substrate motif arginine-arginine-any amino acid-serine/threonine (RRXS*/T*) which is shared by the AGC-family of kinases which includes PKA (Figure 1D lanes 5–8). One of these PKA protein substrates is the nuclear protein CREB. A comparison of different commercially available antibodies against pCREB in SCNCRE cells treated with forskolin or left untreated (control) cells was undertaken before and can also be found in the supplements (Appendix A). Among those antibodies available which show a signal at the expected size(es) and a difference between forskolin and control, we selected the rabbit monoclonal 87G3 from Cell Signaling (used also in Figure 1D lanes 9–12 and Appendix A) for both Western/immunoblot and ICC. After prolonged elevation of cAMP, the catalytic subunit of PKA translocates to the nucleus where CREB is phosphorylated at serine-133 (pCREB) (Figure 1D lane 1 and 2 and Appendix A). An increase in the level of pCREB is also evident at Zeitgeber ZT14 (two hours after lights off) versus (ZT) 10 (two hours before lights off) in extracts of SCN ex vivo (Figure 1D lane 11(ZT10) and 12(ZT14)). Both Western blot of SCNCRE cell extracts and immunocytochemistry (ICC) against pCREB exerted elevated levels of pCREB (Appendix A) with the typical nuclear staining (Appendix A right side) compared to untreated controls (Appendix A left side). 

### 2.3. Elevation of cAMP in SCNCRE Cells

Application of the adenylate cyclase activator forskolin (10 µM) elevated extracellular cAMP levels more than 100-fold over basal (control) levels in two separate experiments performed on a 96-well and a 24-well plate (*p ≤* 0.0001, *n* = 5, ANOVA with Bonferroni post-test; Figure 2) against vehicle controls. Cell culture medium and vehicle control cAMP levels were not significantly different. In an additional experiment, forskolin (1 µM) elevated intracellular cAMP levels significantly only after 30 min and 2 h of forskolin application in comparison to a time-matched control. After 4-, 6- and 12-h cAMP levels were not significantly different between control and forskolin-treated SCNCRE cell cultures (Appendix A).

### 2.4. Characteristics of the CRE-Luciferase Response in SCNCRE Cells

As stated above, SCNCRE cells are stably expressing a CRE-luciferase reporter gene (CRE-luc) and the luminescence output produced by this CRE-luc reporter is displayed in relative luminescence units (RLU). Figure 3A shows that application of forskolin (1µM) to cultures containing 25.000 cells per well of a 96-well plate leads to an elevation of luciferase activity over untreated controls (Figure 3A) which depends on the concentration of luciferin in the cell culture medium. A stable relation between forskolin (1 µM) and control activity is reached at luciferin concentrations above 200 µM. This forskolin-induced elevation of CRE-dependent luciferase activity also depends on the number of cells plated and reaches a stable forskolin-treated to untreated control relation above a cell number of 10.000 cells per well with a luciferin concentration fixed at 500 µM (Figure 3B). Figure 3C shows the dose-dependent increase of CRE-dependent luciferase activity upon forskolin treatment with 25.000 cells per well and a luciferin concentration of 250 µM. Besides cell number plated, luciferin- and forskolin-concentration, the timing of the response onset, maximum and offset of activity are other important parameters to consider. At a constant cell number of SCNCRE cells per well, the maximum response to forskolin (10 µM) was approximately an hour later than the response to forskolin (1 µM), and the maximum of forskolin (0.1 µM) was even earlier (Figure 3D). Thus, the higher the applied concentration of forskolin, the later the maximal response time.

For the following experiments, the number of cells plated out was fixed to 25.000/well and the final luciferin concentration to 250 µM if not indicated otherwise. To evaluate if the observed CRE-luciferase elevation is mediated by the suspected signal transduction proteins, we next tested the influence of the adenylyl cyclase inhibitor MDL12.330A (Figure 4A). MDL12.330 (10 µM) showed significant inhibition of the maximum response of forskolin (1 µM) (Figure 4A). However, MDL12.330 applied alone displayed drastically lower basal RLU values compared to the vehicle controls, which indicates a toxic effect. This presumably toxic effect is also seen in Figure 4B, where forskolin (0.1 µM) is nearly fully inhibited by MDL12.330 but only compared to the vehicle control, not to MDL12.330 (10 µM) alone. One conclusion from these data could be that MDL12.330 at 10 µM is not inhibiting specifically adenylyl cyclase but is most probably toxic to the cells.

The next level of regulation is cAMP-activated protein kinase (PKA). A well-established class of PKA inhibitors are the equatorial diastereomers of 3′,5′-cyclic adenosine phosphorothioate. In particular, the *Rp*-cAMPS analogue, *Rp*-8-Br-cAMPS, has been shown to be an efficient inhibitor of PKA. Figure 5A shows that the SCNCRE cell response to forskolin (1 µM) was inhibited by approximately 50% by *Rp*-8-Br-cAMPS (1 mM), and the response to forskolin (0.1 µM) was nearly abolished (Figure 5B).

Another important protein class relevant to the regulation of CRE-luc activity are the 3′,5′-cyclic adenosine phosphodiesterases (PDE). Many PDEs catalyze the degradation of cAMP to 5′-AMP (or cGMP to 5′-GMP), but some also bind cAMP or cGMP in a separate site. In the presence of a PDE inhibitor in a cell expressing PDEs, the concentration of cAMP rises. A widely used PDE inhibitor exhibiting a broad inhibition spectrum across the 11 described PDE families is isobutyl-methyl-xanthine (IBMX) [19]. Here, the application of IBMX (10 µM) showed robust induction of CRE-luc activity in SCNCRE cells with a maximum after approximately four hours (Figure 6A). When co-incubated with forskolin (1 µM; which peaks later), the resulting activation curve in SCNCRE cells showed an intermediate maximum of CRE-luc activity. The onset was slower than under treatment with IBMX alone and the maximum time was nearly identical to the maximum time of forskolin alone. A similar result was seen with IBMX (10 µM) and forskolin (0.1 µM) combined (Figure 6B).

Another cellular system of potential relevance for cAMP-mediated CRE-luc activity is the organic anion exchanger (OAE). Under the influence of high intracellular cAMP levels (like when cells are incubated with strong cAMP-elevating agents like forskolin or IBMX), OAE enables the release of cAMP from the cell into the extracellular space. A well-established inhibitor for OAE is probenecid, an established medical compound which serves to reduce the excretion of organic ions from the kidney. Probenecid (10 µM) had no significant additional effect on CRE-luc activity in SCNCRE cells when co-applied with forskolin (1 µM) (Figure 7A), however at lower forskolin (0.1 µM) concentration probenecid elevated the reaction to forskolin significantly. However, compared to the impact of IBMX, this approximate 20% further elevation appears to be of rather minor importance.

As mentioned above, besides concentration and mechanisms of action, we observed the timing of the SCNCRE luciferase responses. To further evaluate this, we applied the PKA agonists *Sp*cAMPS and *Sp*cDBIMPS and the EPAC activator (EPAC-A) in addition to forskolin (Figure 8). Next, we compared the onset and maximum activity of CRE-luciferase activity for forskolin (1 µM), *Sp*cAMPS, *Sp*cDBIMPS and EPAC-A (all 100 µM). *Sp*cAMPS displayed both the fastest onset and earliest significant elevation over control levels (*p* < 0.001 over control at 1 h). *Sp*cAMPS is followed by *Sp*cDBIMPS and forskolin. EPAC-A did not significantly elevate SCNCRE luciferase activity at any time point. The CREluc activity increase in response to *Sp*cAMPS application remained the highest until after 6 h of incubation. *Sp*cAMPS, *Sp*cDBIMPS and forskolin reached a similar maximal plateau, at which the RLU values did no longer increase and were not significantly different (Figure 8).

Another class of protein kinases which was shown to be able to phosphorylate CREB and thereby activate pCREB/CRE-dependent transcriptional responses are the protein kinases C (PKC). The classical PKCs (α, β, γ) are activated by a class of substances called phorbol esters. In contrast to forskolin (10 µM), the phorbol ester PMA (1 µM) did not change cAMP levels (Figure 9A). However, PMA did elevate pCREB levels significantly higher than Forskolin and as high as the combination of both PMA and forskolin (Figure 9B) when compared by Western blot analysis. In the CRE reporter gene activity assay after one hour, the RLU levels under PMA treatment did not differ from vehicle control, whereas the forskolin RLU was significantly elevated and the combination of forskolin and PMA (Figure 9C). The complete time course presented in Figure 9D shows that PMA elevated CRE-dependent RLU values with an early and low maximum around 3–4 h after onset, an acceleration of the onset in combination with forskolin and a more than additive maximum (Figure 9D).

## 3. Discussion

The suprachiasmatic nucleus (SCN) of the mouse consists of a number of different cell types [7]. Among the larger neuronal populations of the SCN are cells containing the neuropeptide vasoactive intestinal peptide (VIP) [20]. In the VIP-positive population, some cells apparently do not display circadian expression of clock genes. One theory is that these cells serve as “input sensors” for external stimuli, perceive these and modulate the neighbouring rhythmic cells [13,21].

From the SCN cell line made available by the group of David Earnest, we extracted cells which did not display any detectable luciferase output. These cells were stably transfected with a CRE-luciferase construct (CRE-luc) and further characterized. The clones stably expressing CRE-luciferase and named SCN-CRE (SCNCRE) were strongly VIP-positive. Like the original cell line, they seem to consist of two morphologically different populations, large flat cells rapidly forming a dense monolayer and small, round cells located on top of the large cells forming long protrusions and establishing connections to each other under prolonged cultivation duration [13].

We tested a series of marker proteins to evaluate the neuronal nature of the SCNCRE cells and their developmental stage. Interestingly, although SCNCRE is an immortalized cell line, they expressed in the confluent stage two makers for mature neurons, ßIII-tubulin and “microtubule-associated-protein-2B” (MAP2B), no marker for immature (Nestin) or maturing neurons (Doublecortin). Taken together, SCNCRE cells express neuronal markers suggestive of rather mature neuronal phenotype cells but do not express GFAP.

SCNCRE cells displayed an elevated CRE-luc activity after the application of forskolin, a diterpene secondary metabolite from the Indian foul tree, *Coleus forskolii* [22]. Forskolin is widely used to induce cAMP elevation [23]. In the data presented here, cAMP-synthesis by adenylate cyclase (AC), cAMP-degradation by 3′,5′-phosphodiesterase (PDE) and cAMP-export via organic anion exchangers (OAE) affected CRE-dependent signalling. Besides the various signalling pathway components involved, differential timing of cAMP-production, -degradation and -export processes are important for the latency time of serine^133^pCREB phosphorylation, CRE-dependent luciferase activity onset and maximum reaction, especially when interacting with other parallel pathways like the protein kinase C (PKC) pathway.

Regarding the type of adenylate cyclase present in SCNCRE cells, data are inconclusive. In our experiments, MDL12330 became toxic around the concentrations where it should inhibit AC and SQ22536, another presumed AC inhibitor, did not inhibit forskolin-mediated CREluc activity in SCNCRE cells (Appendix A). Thus, the type and regulation of the AC in SCNCRE cells remain to be investigated.

SCNCRE also reacted to the application of the PDE-resistant cAMP analogues, *Sp*cAMPS and *Sp*cDBIMPS, with *Sp*cAMPS being the faster elevator of CRE-luc activity. This finding may be explained by SCNCRE cells predominantly expressing PKA regulatory subunit isoforms IIα and Iiβ for which *Sp*cAMPS is a better activator than *Sp*cDBIMPS, which displays a strong selectivity for PKA I isoforms [24]. *Sp*cAMPS was also the agent with the fastest CRE-luc response, already elevating levels significantly over basal after 60 min, followed by *Sp*cDBIMPS and forskolin. 

EPAC-A selectively activates the EPAC proteins (EPAC1 and EPAC2, also known as Rap guanine nucleotide exchange factor 3, GEF3) representing a family of intracellular sensors for cAMP, which function as nucleotide exchange factors for the Rap subfamily of RAS-like small GTPase [25]. Here, the EPAC-activator (EPAC-A) did not exert any significant change in CRE-luc activity compared to controls. EPAC-A does not activate PKA, nor does it significantly influence AC, PDE or OAE activity [25]. We conclude that EPAC plays no significant role in CRE-luciferase activity modulation.

Another finding of relevance is the shift of onset and the estimated time point of maximum when agents were combined. The onset of CREluc activity in SCNCRE cells is faster for IBMX than for forskolin. When both are applied in combination the onset of activity is not significantly slower than that of IBMX alone. However, the maximal response of the combination is at a time point similar to the maximal time response to the forskolin application. Thus, the combination response behaves similar to the faster substance (IBMX) at the onset of activity and similar to the slower substance (forskolin) at the maximum time. The effect that the higher the dose of forskolin, the later the maximal response time also applies for the combination of forskolin with IBMX in that forskolin determines at what exact time point this maximum occurs. Thus, our data support the interpretation that the extent of the temporal shift of the maximum time depends on the dose of forskolin.

Beside PKA, several other protein kinases have been shown to be able to phosphorylate CREB at serine-133 [26]. One of these “CREB- kinases” is protein kinase C (PKC) which represents a family of fifteen isozymes in humans [27]. The classical PKCs (α, β, γ) can be activated by a class of substances called phorbol esters, like the cell-permeable substance phorbol-12-myristat-13-acetat (PMA) [28]. PKCs are known to activate other protein kinases like the p42/44-MAP-kinases, which can also be present inside the cell nucleus or act via shuttle systems like the one involving pp90RSK [29]. Since PMA and forskolin combination is neither simply additive nor synergistic or antagonistic on CRE-dependent transcription activity, the kinases involved in the PMA-activated pathway at least partly compete with those involved in the forskolin-induced pathway. 

In contrast to forskolin, PMA did not modulate cAMP levels in SCNCRE cells in our experiments. However, PMA did elevate pCREB levels after one-hour incubation, as shown by Western blot analysis. Under PMA treatment, pCREB levels were higher than under forskolin treatment and as high as the combination of both PMA and forskolin. However, after one-hour CRE-luc activity under PMA treatment did not differ from vehicle control but rose significantly over vehicle control levels after approximately two hours, whereas forskolin significantly elevated CRE-luc activity already after one hour. Thus, neither cAMP levels nor CREB phosphorylation at serine 133 after one hour is a reliable marker for the maximum or the timing of maximal induction of CRE-dependent transcriptional activity in SCNCRE cells.

It is widely assumed that in the SCN in vivo, different populations of parvocellular neurons located in different regions exert very different tasks with respect to rhythm generation, as well as input and output signalling [8,9,10,11,30,31,32,33,34,35]. We used SCNCRE cells, a VIP-expressing transformed cell population derived from the mouse SCN, to investigate the factors determining the cAMP/PKA/CRE-mediated transcriptional response of such cells. 

Our conclusion from the experimental data presented here is that there is a dose- and time-sensitive cAMP-mediated transcriptional response that is mostly depending on AC-activation and PDE-expression and -inhibition and, to a lesser degree and only at lower stimulant concentration, on the egress of cAMP through OAG. This cAMP-dependent pathway can interact with PKC-signalling but does not or only marginally depend on other cAMP-target proteins like EPAC. 

For the SCN in situ, these data suggest a strong coupling and interaction of the different neuronal cell populations of which those that are non-rhythmic should not be neglected regarding their potential influence on input- and phase-regulation and output functions of other cell populations inside the SCN.

## 4. Materials and Methods

### 4.1. Cell Culture

Immortalized mouse suprachiasmatic nucleus cells (200.000 per well in a 6-well plate; kindly supplied by Prof. David J. Earnest PhD, Texas A&M University, Bryan, TX, USA [13] devoid of spontaneous (or basal) luminescence activity were cultivated in Dulbecco’s Modified Eagle’s Medium (DMEM; Sigma, Deisenhofen, Germany) supplied with 10% fetal bovine serum (FBS), penicillin/streptomycin (100 U/mL) and GlutaMax (Life Technologies, Darmstadt, Germany). This preparation is from here on called complete medium, while the preparation without FBS is called serum-free medium. 

These cells were transfected using FuGene HD (Roche, Mannheim, Germany) and a commercially available CRE-luciferase plasmid (pGL4.29[luc2P/CRE/Hygro]) from Promega (Heidelberg, Germany). After maintaining the cells with the transfection reagents overnight, the medium was supplemented with Hygromycin (250 µg/mL) and left for at least three days. The cells were then washed three times with sterile Hank’s Balanced Salt Solution (HBSS; Gibco, Life Technologies, Darmstadt, Germany). Those cells still attached to the cell culture plate were enzymatically detached from the cell culture substrate using Accutase-Solution (Accutase; Sigma, Deisenhofen, Germany), counted (manually or by using Scepter 2.0, Merck-Millipore, Darmstadt, Germany) and supplied with medium containing the antibiotic selection agent Hygromycin (50 µg/mL; Enzo Lifescience GmbH, Lörrach, Germany). The surviving (Hygromycin-resistant therefore CRE-reporter-positive) cells were plated out for a selection of stably transfected clones. Mixed clones showing stable induction of CRE-luc-activity induced by forskolin application (10 µM; more than four-fold elevated luminescence in the presence of forskolin over vehicle control (1% DMSO)) were expanded and frozen at a density of 1.5 million/vial at −80 °C in freezing medium (IBIDI, Planegg, Germany). The resulting cell line was named SCNCRE. 

SCNCRE cells thawed from these preparations were given passage number 1 and used up to passage 13 without detectable changes in control experiments comparing forskolin-treated with untreated cells. Cells were passaged once a week at a density of 1 million cells per 75 cm^2^ flask.

### 4.2. Western Blot Analysis

Western blot (immunoblot) analysis was performed essentially as described [36]. In short, 200.000 SCNCRE cells per well in a 24-well plate were seeded in 400 µL complete medium per well and left overnight. The next day, the complete medium was changed to a serum-free medium. At the end of each experiment, serum-free medium was removed, and the cells were immediately lysed in lithium dodecylsulphate sample buffer (2xLDS; Invitrogen/Thermo Fisher, Darmstadt, Germany). Cell extracts were sonified five times for 3 s using a 24 kHz ultrasound sonifier (Dr. Hielscher, Teltow, Germany), heated for 10 min at 70 °C, chilled on ice and centrifuged for 5 min at 13.000 rpm in an Eppendorf cap centrifuge (Eppendorf, Hamburg, Germany). Samples were either used immediately for gel electrophoretic separation on Bis/Tris gels (NuPage 4–12% gradient gels, MES buffer, Invitrogen/Thermo Fisher, Darmstadt, Germany) or stored at −20 °C. After the electrophoretic run, the gels were blotted on PVDF membrane using iBlot (Invitrogen/Thermo Fisher, Darmstadt, Germany) washed with Tris-buffered saline (pH 7.6) containing 0.1% (vol/vol) Tween-20 (TBS-T; Sigma, Deisenhofen, Germany), blocked for 1 h at room temperature using Rotiblock (Roth, Karlsruhe, Germany) and incubated with primary antibodies diluted in Rotiblock overnight at 4 °C [16,36]. The antibodies used for Western blot were rabbit anti-VIP, rabbit anti-PKA regulatory subunit RIα, mouse anti-PKA regulatory subunit RIIα, mouse anti-PKA regulatory subunit RIIβ (Transduction labs), rabbit anti-phospho-PKA substrate (100G7E; rabbit monoclonal; 1:5.000; Cell Signaling Technology, Bad Nauheim, Germany, anti-phospho-CREB (87G3, rabbit monoclonal; 1:1.000; Cell Signaling Technology, Bad Nauheim, Germany), rabbit anti-pro-VIP, anti-Nestin, anti-GFAP, anti-β-III-tubulin, mouse anti-MAP2 and mouse anti-β-actin (1:20.000; Sigma, Deisenhofen, Germany). For a complete list of the antibodies used see Appendix A.

After incubation with the first antibody, the membranes were washed four times for 2 min with TBS-T and incubated with the appropriate secondary HRP-coupled antibodies against rabbit (1:50.000; Santa Cruz Biotechnology, Heidelberg, Germany) or mouse (1:50.000; DAKO, Hamburg, Germany) in Rotiblock for 1 h at room temperature. Membranes were washed with TBS-T four times for 2 min and once for 5 min. Signal detection was performed using the chemiluminescent substrate Luminata forte (Millipore, Darmstadt, Germany) or AceGlow^TM^ Ultrasensitive (PeqLab, Munich, Germany) and a CCD camera- equipped luminescence analysis system (Quantity One, ChemiDoc XRS, Bio-Rad, Hercules, CA, USA; or a Fusion XL system from PeqLab, Munich, Germany).

### 4.3. Immunocytochemistry

5.000 or 25.000 SCNCRE cells were plated on poly-L-lysine-coated glass coverslips (10 mm diameter) and treated with 10 µM forskolin or left untreated (control/1% DMSO) for one hour. For fixation, the medium was removed at the end of each experiment and replaced with phosphate buffer (0.1 M) containing 4% (*w/v*) paraformaldehyde (Morphisto, Frankfurt, Germany) for at least 15 min at room temperature. Samples were washed with 0.1 M phosphate-buffered saline (PBS) for 15 min and blocked for 45 min at room temperature using 5% (*v/v*) normal goat serum (both from Sigma-Aldrich, Darmstadt, Germany) PBS containing 0.3% (*v*/*v*) Triton-X-100 (blocking buffer). The cells were incubated overnight with primary antibodies in a blocking buffer (Benz et al., 2010). The primary antibodies used were anti-β-actin (1:10.000; Sigma-Aldrich, Deisenhofen, Germany) and anti-phospho-CREB (1:500) (Cell Signaling, Bad Nauheim, Germany). Diaminobenzidin (DAB) staining was performed according to a standard protocol as described [37,38]. For fluorescence staining, samples were washed three times for 10 min in PBS and incubated in fluorochrome-coupled secondary antibodies against rabbit (Alexa-568; 1:2.500, Invitrogen) and mouse (Alexa-488; 1:500; Active Motif, Rixensart, Belgium) in blocking buffer. Coverslips were washed three times for 10 min in PBS, mounted on slides using VectaShield HardSet Antifade mounting medium with DAPI (Vector Laboratories, Newark, CA, USA) and stored at 4 °C. Intracellular distribution and fluorescence intensity of immunolabeled proteins was determined by fluorescence microscopy (Axio, Zeiss, Göttingen, Germany).

### 4.4. cAMP ELISA

Extracellular cAMP was determined in SCNCRE cells using serum-free, undiluted cell culture supernatant and a commercial ELISA (Cayman, Ann Arbor, MI, USA; Item no. 581001) according to the supplier’s protocol as described before [36]. In brief, SCNCRE cells were plated either at 25.000 cells per well on a 96-well plate or 400.000 cells per well on a 24-well plate, left overnight in serum-free medium and exposed with the indicated substances for one hour. Subsequently, the medium was harvested and either frozen at −20 °C for later processing or directly subjected to the sample plate provided in the cAMP ELISA kit. The cAMP ELISA procedure was performed according to the supplier’s manual, as previously described [36].

### 4.5. Determination of Luminescence Activity

If not indicated otherwise, 25.000 cells per well of a 96-well plate were plated in 100 µL volume and left in the incubator overnight in order to promote adhesion. Experiments were performed in 200 µL medium/well containing 0.25 mM Luciferin (unless indicated otherwise) and measured in a luminometer (BMG Lumistar; BMG, Ortenberg, Germany) or Berthold Centro LB960 (Berthold Technologies, Bad Wildbad, Germany) at 35° Celsius for 0.1 s per well. Luminescence data are displayed as relative luminescence units (RLU). Cells were measured after different times of incubation (e.g., every 15 min) with and without chemical agents potentially influencing CRE-luciferase signalling unless indicated otherwise.

### 4.6. Image Analysis and Statistics

Images of protein bands were digitized either using a Bio-Rad Universal Hood equipped with a CCD camera, Quantity One and ChemiDoc XRS software (BioRad, München, Germany) or a Fusion system (Vilber/PeqLab; PeqLab/VWR, Erlangen, Germany). Signal intensities of the digitized images were analyzed using online open access ImageJ software (https://imagej.net, accessed on 27 July 2022) [39]. Each area/density value for a specific protein band was normalized against the corresponding β-actin signal of each extract. Statistical significance between groups was analyzed using one-way ANOVA, followed by the Tukey post-hoc test. The criterion of significance was *p ≤* 0.05, with analysis performed using GraphPad Prism 8.4 (GraphPad, San Diego, CA, USA). 

### 4.7. Chemicals/Stimulants

Forskolin (Cat. No. F3917 Sigma-Aldrich) [22], Probenecid (Cat. No. P8761 Sigma-Aldrich) [40], Phorbol 12-myristate 13-acetate (PMA) (Cat. No. P1585 Sigma-Aldrich) [28,41], isobutyl-methyl-xanthene (IBMX; Cat. No. I7018 Sigma-Aldrich) and MDL-12,330A [42,43] (M182 Sigma-Aldrich) were from Merck (Deisenhofen, Germany). *Rp-*8-Br-cAMPS [14,44], *Sp*cAMPS, *Sp*cDBIMPS, EPAC-activator (all from BioLog LifeScience Institute, Bremen, Germany) and SQ22536 (Selleck Chemicals GmbH, Planegg, Germany were dissolved in water. Luciferin (Promega, Heidelberg, Germany) was dissolved directly in the Leibovitz L-15 culture medium at the indicated concentrations (usually 0.5 mM). Reagents or appropriate vehicles were applied to the media for the indicated periods.

### 4.8. SCN Ex Vivo Sample Treatment

Mice were held under a schedule of 12 h of darkness and 12 h of light with unlimited food and water supply according to the veterinary laws of Germany. Animals were sacrificed at Zeitgeber time (ZT)10 (ten hours after lights on) and ZT14 (2 h after lights off. The brains were excised, and the SCNs were punched out with a hollow needle. The punches were immediately mixed with two times concentrated sample buffer homogenized by sonification and treated as described above for the SCNCRE cell extracts for Western blot analysis.

## Figures and Tables

**Figure 1 ijms-23-12226-f001:**
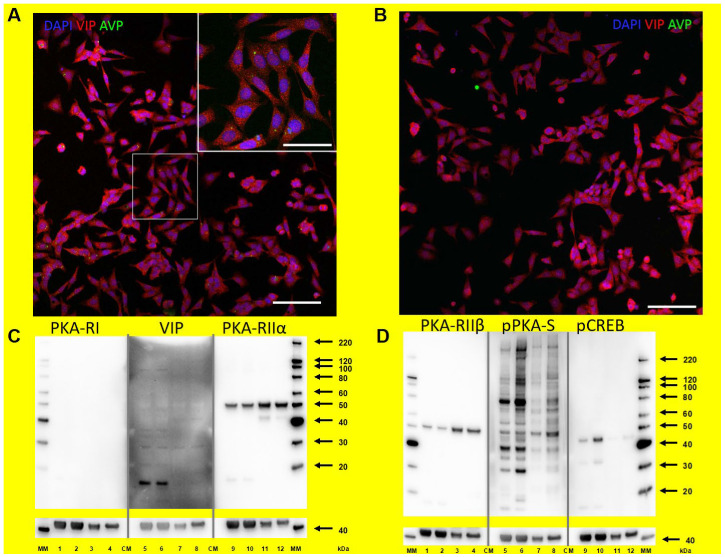
(**A**,**B**): Immunocytochemical staining shows that most SCNCRE cells were vasoactive intestinal peptide (VIP)-positive (red fluorescence), whereas only some cells displayed arginine-vasopressin (AVP)-positive granules (green). Scalebar is 20 µm in the insert and 100 µm at the lower magnification pictures. (**C**): Protein extract (lanes 1, 5, 9: control SCNCRE cells, lanes 2, 6, 10: forskolin-treated (10 µM), lanes 3, 7, 11: SCN ex vivo at ZT10, lanes 4, 8, 12: SCN ex vivo ZT14) were tested for immunoreactivity against different antibodies. Lanes 1–4 anti-PKA regulatory subunit type Iα/β (PKA-RI; no signal), lanes 5–8 anti-pro-VIP (VIP; 17 kDa signal) and lanes 9–12 anti-PKA regulatory subunit type IIα (PKA-RIIα; 52 kDa signal). Anti-β-actin (41 kDa) is shown as a loading control for all lanes. (**D**): Whole protein extract (lanes 1, 5, 9: control SCNCRE cells, lanes 2, 6, 10: forskolin (10 µM), lanes 3, 7, 11: SCN ex vivo ZT10, lanes 4, 8, 12: SCN ex vivo ZT14) were tested for immunoreactivity against different antibodies. Lanes 1–4 anti-PKA regulatory subunit type IIβ (PKA-RIIβ; 54 kDa signal), lanes 5–8 anti-phospho-PKA substrate (pPKA-S) and lanes 9–12 anti-serine-133-phosphorylated CREB (pCREB) (41 kDa signal). Anti-β-actin (40 kDa) is shown as a loading control for all lanes.

**Figure 2 ijms-23-12226-f002:**
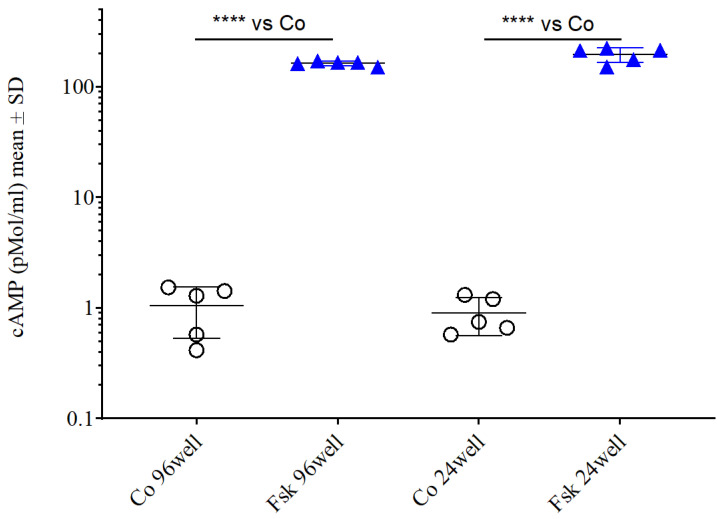
Extracellular cAMP levels in SCNCRE cells after treatment with forskolin (Fsk; 10 µM; blue triangles). Extracellular cAMP levels were elevated significantly by 10 µM forskolin (ANOVA with Bonferroni post-test; *p* < 0.00001; ****; *n* = 5) in both experiments (Exp 1; 96 well plate; Exp2; 24 well plate). Both control (circles) and the forskolin-treated cells under both conditions were not significantly different (Shown are the means values ± SD; ANOVA with Bonferroni post-test; n.s.; *n* = 5).

**Figure 3 ijms-23-12226-f003:**
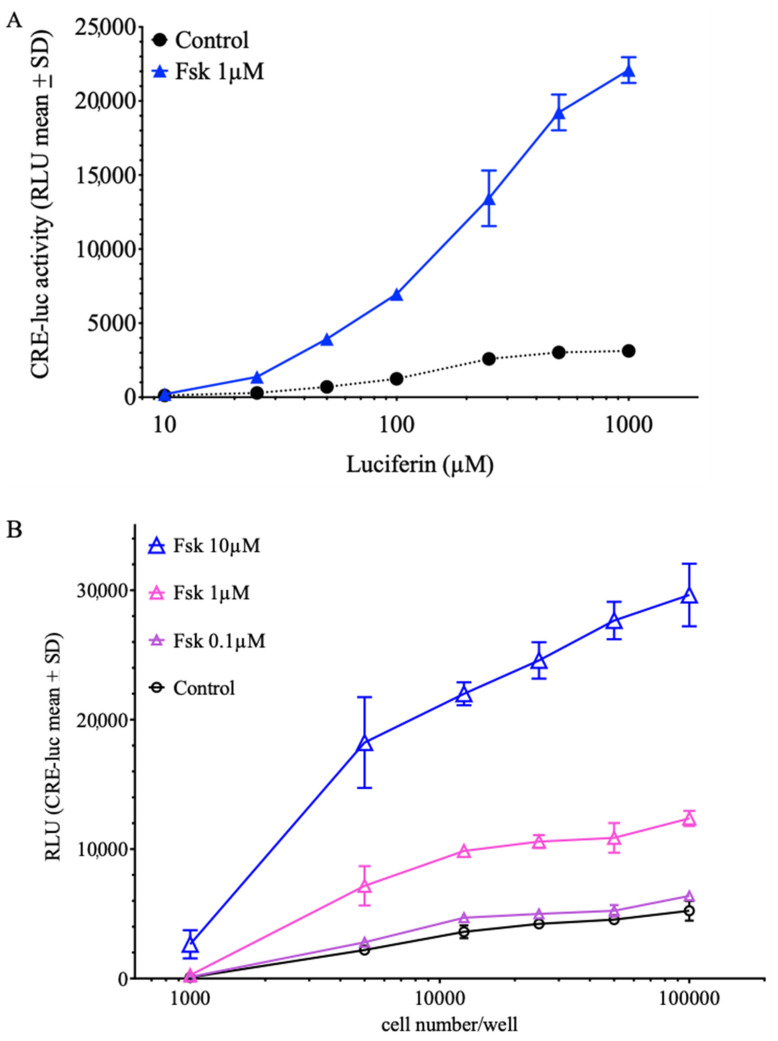
CRE-luciferase activity variation in SCNCRE cells under different experimental conditions. (**A**) Influence of luciferin concentration in the medium at a constant number of cells (25.000/well) left untreated or treated with 1 µM forskolin (Fsk). A stable relation between control and forskolin-treated RLUs was reached at a concentration of 200 µM luciferin. (**B**) Influence of the number of cells seeded on the CRE-luciferase activity of forskolin-treated and untreated control cells. (**C**) Influence of forskolin after one hour of treatment at 25.000 cells per well and a final concentration of 250 µM luciferin. (**D**). With a constant number of cells plated (50.000/well) and 500 µM luciferin in the medium, the maximal response to 10 µM forskolin was detected one hour later than that to 1 µM forskolin and 2 h later than the elevation under 0.1 µM and control. (Shown are the mean values ± SD; ANOVA with Bonferroni post-test; *n* = 5).

**Figure 4 ijms-23-12226-f004:**
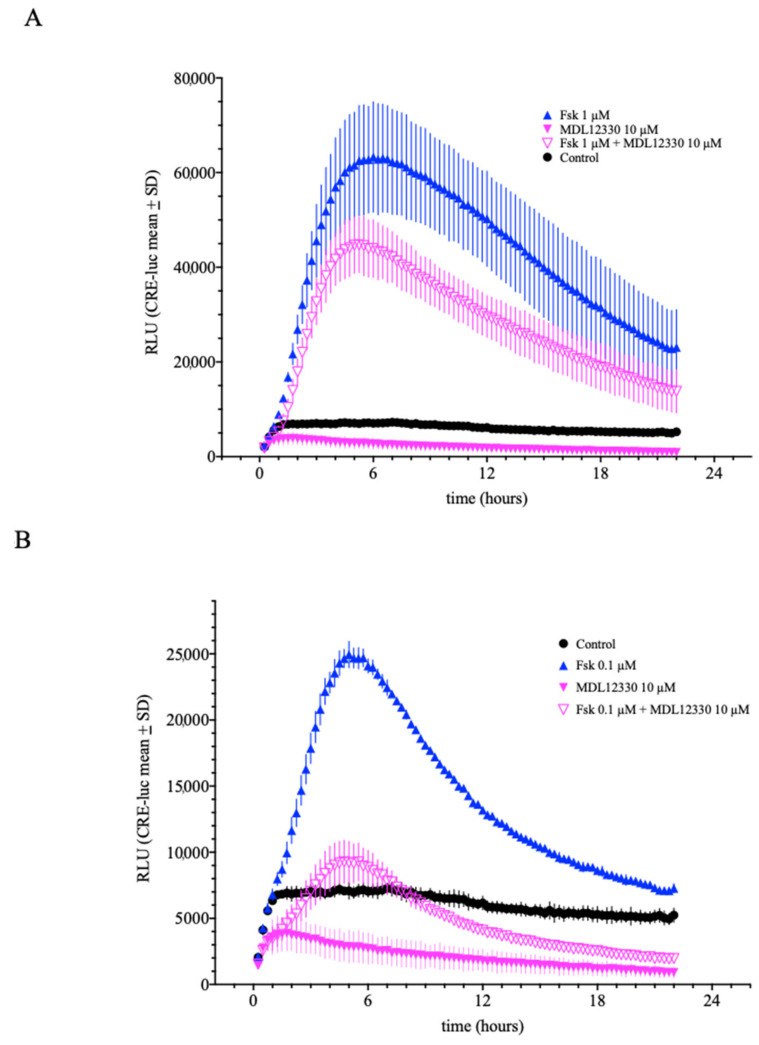
Influence of the adenylyl cyclase inhibitor MDL12.330 on forskolin-induced CRE-luciferase activity. MDL12.330 inhibited forskolin-induced (1 µM and 0.1 µM) CRE-luc activity (**A**,**B**). However, even at 10 µM, MDL12.330 alone, or in combination with forskolin, exerted strongly reduced basal luminescence values in comparison to untreated control cells indicating cell toxicity. (Shown are the means values ± SD; ANOVA with Bonferroni post-test; *n* = 4).

**Figure 5 ijms-23-12226-f005:**
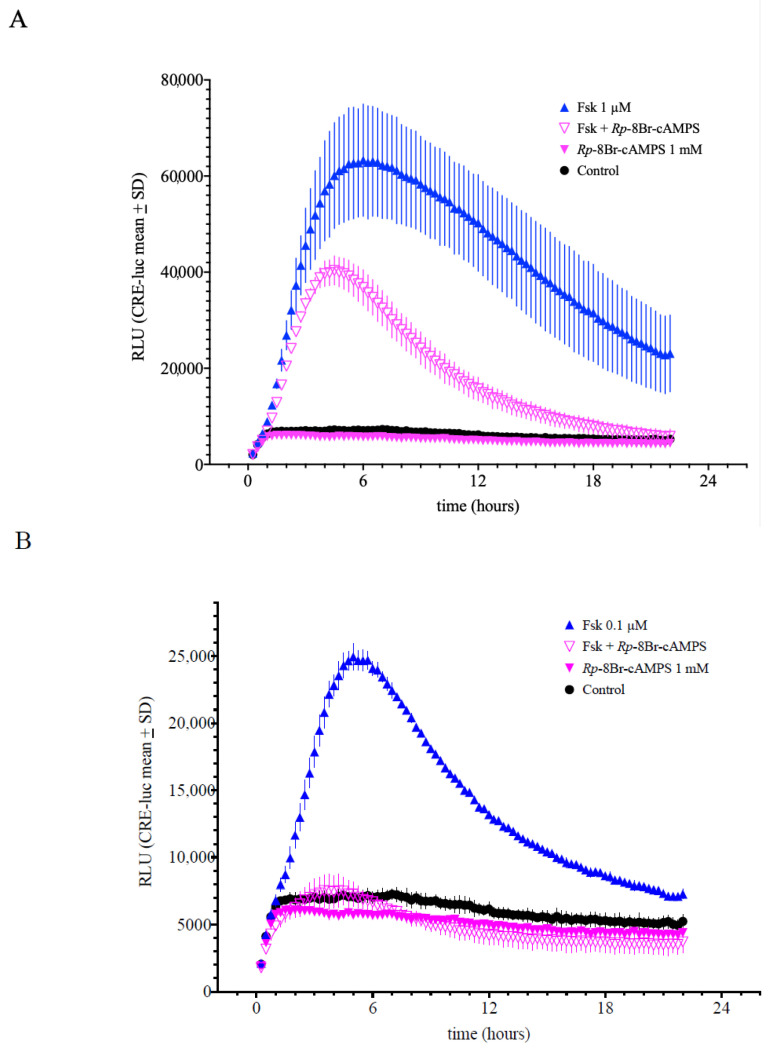
Influence of the protein kinase A (cAMP-dependent protein kinase = PKA) inhibitor *Rp*-8-Br-cAMPS on forskolin-induced CRE-luciferase activity. *Rp*-8-BrcAMPS inhibited CRE-luciferase activity induced by 1 µM and 0.1 µM forskolin (**A**,**B**). *Rp*-8-BrcAMPS displayed significant inhibition if applied in parallel to forskolin. *Rp*-8-BrcAMPS alone exerted similar luminescence values compared to untreated control cells indicating no toxic side effects. (Shown are the means values ± SD; ANOVA with Bonferroni post-test; *n* = 5).

**Figure 6 ijms-23-12226-f006:**
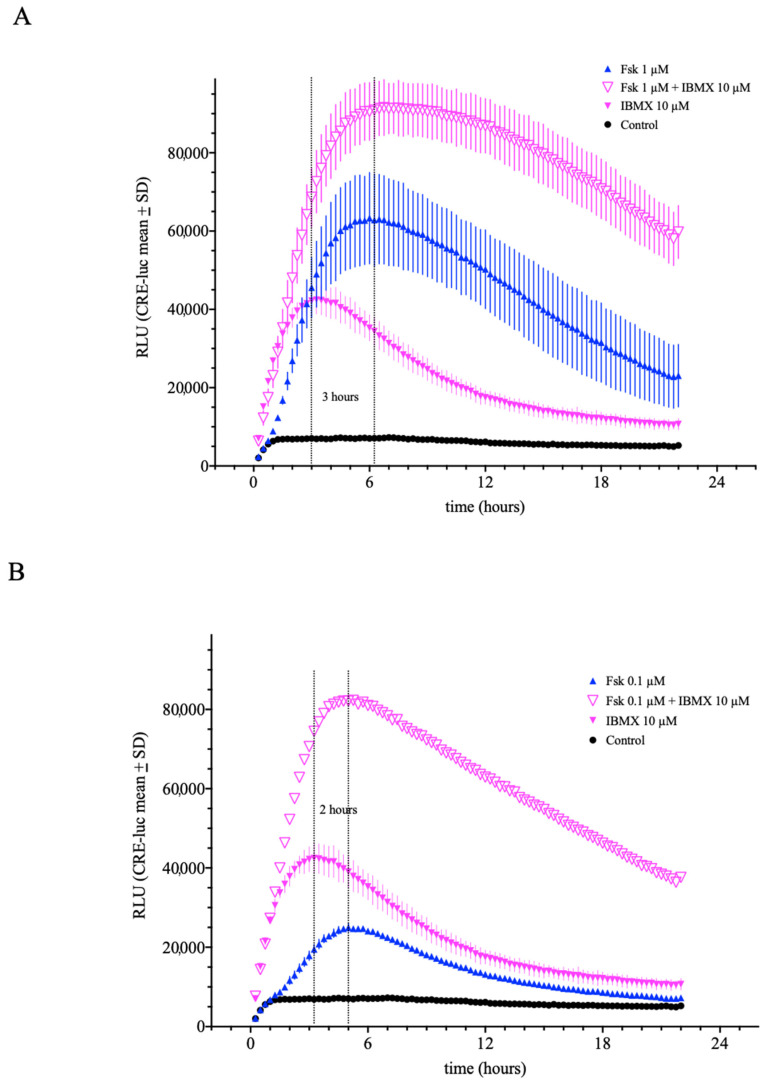
Influence of the non-specific phosphodiesterase inhibitor isobutyl-methyl-xanthene (IBMX) on forskolin-induced CRE-luciferase activity in SCNCRE cells. IBMX elevated CRE-luciferase activity induced by 1 µM and 0.1 µM forskolin (**A**,**B**). IBMX application displayed significant activation if applied alone which is further elevated when co-applied with forskolin. IBMX displays an earlier maximal response compared to forskolin (approximately 2 h earlier). (Shown are the means values ± SD; ANOVA with Bonferroni post-test; *n* = 5).

**Figure 7 ijms-23-12226-f007:**
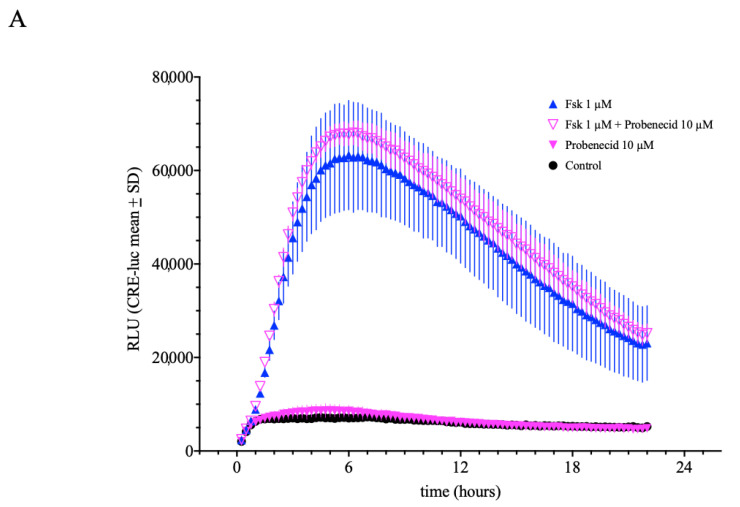
Influence of the organic anion exchanger (OAE) inhibitor probenecid on forskolin-induced CRE-luciferase activity. Probenecid elevated CRE-luc activity induced by 1 µM forskolin (**B**) but not at 0.1 µM (**A**). Probenecid application alone had no significant effect indicating a minor influence of cAMP export on forskolin-induced CRE-luciferase activity. (Shown are the means values ± SD; ANOVA with Bonferroni post-test; *n* = 5).

**Figure 8 ijms-23-12226-f008:**
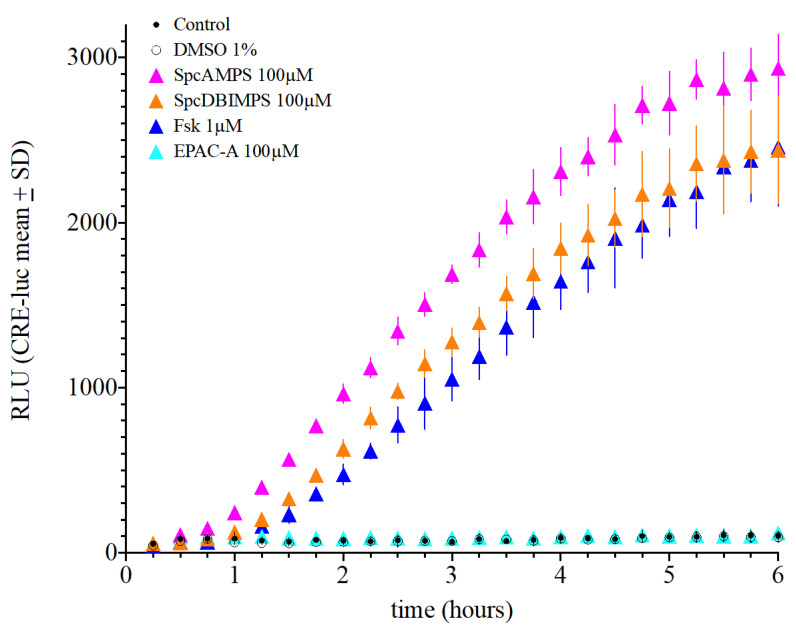
Comparison of the onset and maximum activity of CRE-luciferase activity (in RLU) for forskolin, *Sp*cAMPS, *Sp*cDBIMPS and EPAC-Activator in comparison to control. Note that *Sp*cAMPS displays the fastest onset (*p* < 0.001 over control at 1 h), followed by *Sp*cDBIMPS and, forskolin. EPAC-A did not significantly elevate SCNCRE luciferase activity at any time point. (Shown are the means values ± SD; ANOVA with Bonferroni post-test; *n* = 4–8).

**Figure 9 ijms-23-12226-f009:**
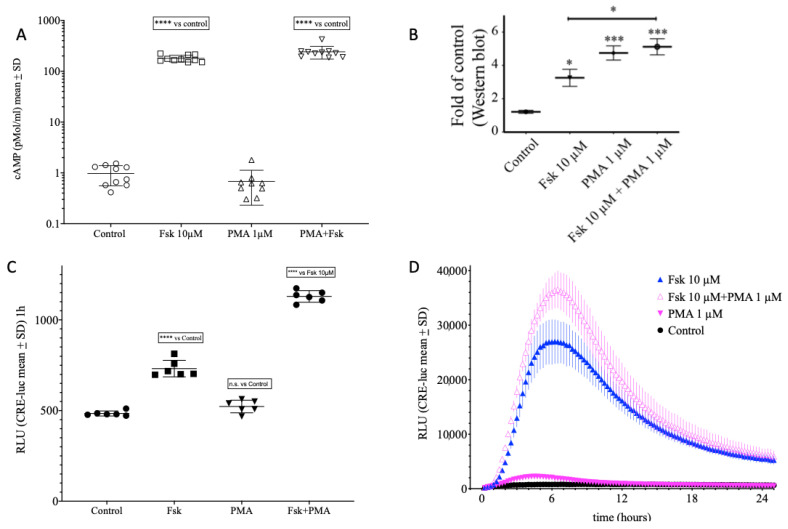
Combined effect of PKA activation by forskolin and protein kinase C (PKC) activation by phorbol-myristate-acetate (PMA). (**A**) extracellular cAMP levels of SCNCRE cell left untreated (control) or treated with 10 µM forskolin (Fsk), 1 µM PMA, or 10 µM Fsk plus 1 µM PMA. (**B**) β-Actin normalized Western blot area values of a SCNCRE cells left untreated or treated with 10 µM forskolin (Fsk), 1 µM PMA or 10 µM Fsk and 1 µM PMA treated for one hour and probed with an antibody against the serine 133 phosphorylated form of CREB (pCREBS133). The original blot images for pCREB and β-Actin are displayed in Appendix A. (**C**) CRE-luciferase activity after 1 h. Note that Fsk and Fsk + PMA are significantly higher than control (*p* < 0.0001; ****) and Fsk + PMA is significantly higher than Fsk (*p* < 0.05; *) but in contrast to the Western blot, PMA had no significant effect on its own. (**D**) Full time course of levels of SCNCRE cell left untreated or treated with 10 µM forskolin (Fsk), PMA, or Fsk plus PMA. Note that in contrast to cAMP, PMA elevates SCNCRE luciferase activity, acts additive with forskolin without affecting cAMP levels and with a profound effect on CREB phosphorylation at serine-133 not significantly different from the combination with forskolin. Neither cAMP levels nor CREB values accurately predict the effect on CRE-mediated transcriptional activity. (Shown are the means values ± SD; ANOVA with Bonferroni post-test; *n* = 4–8; *p*< 0.001 = ***; n.s. = not significant).

## Data Availability

All raw data and calculations are available from the senior author.

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
