# Peer review of "Regulation of CRE-Dependent Transcriptional Activity in a Mouse Suprachiasmatic Nucleus Cell Line"

_ijms, 2022, doi:10.3390/ijms232012226_

Round 1

Reviewer 1 Report

The authors studied the CRE-dependent transcriptional activity in a mouse suprachiasmatic cell line after different pharmacological treatments related to cGMP regulation or protein kinases. The paper is well written and the results are convincing.

However, I feel that the major drawback of this paper is that the novelty is not high. Pharmacological treatments applied are classical but no surprising outcomes were obtained. I think the authors can improve the paper quality by going a bit into detail through more cGMP assays. For example to examine if the peak of RLU overlap or lags behind the peak of cGMP by measuring intracellular cGMP conc., if the probenecid could reduce the extracellular cGMP conc., and etc. . 

Minor points are that the chart and text size of the figures do not seem proper and can be improved.

Based on the above, I suggest the authors consider the possibility to perform more cGMP assays. The paper can be accepted after that.

Author Response

Reviewer comments in italics and our response in red.

Reviewer 1

The authors studied the CRE-dependent transcriptional activity in a mouse suprachiasmatic cell line after different pharmacological treatments related to cGMP regulation or protein kinases. The paper is well written and the results are convincing.

However, I feel that the major drawback of this paper is that the novelty is not high. Pharmacological treatments applied are classical but no surprising outcomes were obtained. I think the authors can improve the paper quality by going a bit into detail through more cGMP assays. For example to examine if the peak of RLU overlap or lags behind the peak of cGMP by measuring intracellular cGMP conc., if the probenecid could reduce the extracellular cGMP conc., and etc. . 

Minor points are that the chart and text size of the figures do not seem proper and can be improved.

Based on the above, I suggest the authors consider the possibility to perform more cGMP assays. The paper can be accepted after that.

We thank reviewer 1 for the overall positive comment on our manuscript! We agree that at first glance some of the results appear to lack novelty and made some preliminary experiments on the requested relation between cAMP levels and the CRE-reporter gene output. We have also added new data on intra- and extracellular cAMP levels which provide preliminary evidence for a spike of cAMP elevation that is visible and significant at 30 minutes and 2 hours of stimulation with forskolin (1µM), but after 4, 6 and 12 hours control and forskolin levels do not differ any more (Suppl. Fig.7).

The mentioned chart and text size issues have been corrected.

Reviewer 2 Report

The paper by Monika Langiu et al. deals with the activation of cAMP/CREB activity in cells regulating the central circadian clock in mammals.

For this, the authors designed a CRE-luciferase  read-out next to the CRE-binding element to analyze the cAMP-pathways in these cells.

It´s a need idea but I´m wondering about the novelity of this study. What´s new? EPAC? Please specify.

Overall, there are several inaccuracies in the Figures and Figure legends.

Major remarks

I`m missing a few sentences about circadian rhythm and cAMP overall in the introduction and an explanation why the authors used these specific time points Z10 and Z14. What´s new?  What´s the mechanism by which PMA  induces CREB phosphorylation (discussion).

Figure 1: Figure B is redundant as the data are already shown in A.  It is not stated that this are merged figures. In C and D it will be helpful to include the antibodies in the top of the gels. ZT10 and ZT14 are not explained. In D, there is no pCREB visible in lanes 11 and 12 as stated in paragraph 3.2.

In this respect, the  ex ante validation of several pCREB antibodies is to point out positive.

Why did the authors first optimized the luciferin concentration to 250 µM and then used 500 µM in Fig. 3D?

Fig. 4: For these experiments, the authors used  MDL12.330A, a AC inhibitor,  and detected a toxic effect on the cells. The experiments should  be repeated with another inhibitor (SQ22536) to unambiguously analyze the effect of AC  in these cells.

Figure 9B: Please, include the blot to visualize the data points.

Minor remarks

Methods Western Blot: How long are the cells kept in serum-free medium before stimulation?

Figures 2 and 9A,C. Please use lines as in 9B to illustrate significances.

Figure 2: control 24 well

Figure 3B/D: The use of similar symbols is very unfavorable. Pleas use different sybols (open and closed and/or colours (see Figure 8). Same holds for 4A,B control and MDLI symbols; 5A;B control and Rp; 6A,B control and IBMX; 7A,B control and probenecid as well as 9D with control and Fsk.

Figure 3A,C: labeling of Y-axis differs from all other figures.

Figure 3C: Forskolin concentration in the insert.

Figure 3D: what is the meaning of the “k” in the inserted figure legend ?

Figure 3 legend: B) Influence of cell number seeded on forskolin-treated and untreated (control) cells. C) Dose-dependent influence of forskolin …. (please ask a native speaker!)

Figure 4 and 5 legend: … by  1 µM (A) and 0.1 mM (B) …… Next sentence “Rp-8-Br-cAMPs displayed…” is done twice.

Figure 9, row 4: -- values of a SCNCRE ….

Author Response

The reviewer report is in italics and our response in red

Reviewer 2

The paper by Monika Langiu et al. deals with the activation of cAMP/CREB activity in cells regulating the central circadian clock in mammals.For this, the authors designed a CRE-luciferase  read-out next to the CRE-binding element to analyze the cAMP-pathways in these cells.It´s a need idea but I´m wondering about the novelity of this study. What´s new? EPAC? Please specify.

The novelty aspect of our work is first and foremost in its time-resolved analyses. Most studies do not provide such detailed information on onset, maxima and offset of transcriptional responses to cAMP alone and in interaction with other signaling pathways. Secondly, as reviewer 2 states correctly, the involvement of EPAC (or cyclic nucleotide dependent ion channels) in SCN cAMP signaling has not been ruled out convincingly before. We have also added new data on intra- and extracellular cAMP levels which provide preliminary evidence for a spike of cAMP elevation that is visible and significant at 30 minutes and 2 hours of stimulation with forskolin (1µM), but after 4, 6 and 12 hours control and forskolin levels do not differ any more (Suppl. Fig.7).

Overall, there are several inaccuracies in the Figures and Figure legends.

 The inaccuracies in the Figures and Figure legends have been corrected.

Major remarks

I`m missing a few sentences about circadian rhythm and cAMP overall in the introduction and an explanation why the authors used these specific time points Z10 and Z14. What´s new?  

We agree with reviewer 2 that we have not been explicit enough in regard to the description of circadian rhythms nomenclature for the SCN ex vivo data. In the mouse SCN in situ cAMP levels vary in a diurnal and circadian manner [1]. The term “Zeitgeber” is an established term in the chronobiology field. Zeitgeber time zero (ZT00) described the begin of a light phase in a 12 hour light, 12 hour dark 24 hour long phase setting. Under these conditions (12 hours light, 12 hours dark) the SCN in mice in situ displays low cAMP levels at the end of the light phase (ZT10) and elevated cAMP levels 2 hours after lights off (ZT14). This information has been added to the M&M section 2.8.

The cell line we use here is not spontaneously rhythmic in a diurnal/circadian manner as are several subpopulations of SCN neuronal cells in vivo.

What´s the mechanism by which PMA  induces CREB phosphorylation (discussion).

As stated in the manuscript PMA does not act via elevation of cAMP. However, PMA activates a number of potential CREB-kinases in a direct or indirect way. Protein kinases capable of phosphorylating CREB and activated by PMA are the classical PKCs (α,β,γ). PKCs are known to activate other protein kinases like the p42/44-MAP-kinases which can also be present inside the cell nucleus or act via shuttle systems like the one involving pp90RSK [2]. Since PMA and Forskolin combination is neither simply additive nor synergistic or antagonistic on CRE-dependent transcription activity the kinases involved in the PMA activated pathway at least partly compete with those involved in the Forskolin-induced pathway.

Figure 1: Figure B is redundant as the data are already shown in A.  It is not stated that this are merged figures. In C and D it will be helpful to include the antibodies in the top of the gels. ZT10 and ZT14 are not explained. In D, there is no pCREB visible in lanes 11 and 12 as stated in paragraph 3.2.

We agree that Fig 1 A and B are abundant, but wanted to show two different cultures displaying very similar staining. Antigens stained are now stated above the respective blot lanes. ZT will be explained as stated above. In D12 a very faint band for pCREB is visible as stated in the text. Note that no contrasting or image manipulation was used in lanes 11 and 12 in comparison to lanes 09 and 10.

In this respect, the  ex ante validation of several pCREB antibodies is to point out positive.

Thank you very much!

Why did the authors first optimized the luciferin concentration to 250 µM and then used 500 µM in Fig. 3D?

This was stated wrongly (and has been corrected), in all experiments shown final luciferin concentration was 250 µM.

Fig. 4: For these experiments, the authors used MDL12.330A, a AC inhibitor,  and detected a toxic effect on the cells. The experiments should be repeated with another inhibitor (SQ22536) to unambiguously analyze the effect of AC in these cells.

We ordered SQ22536 and made several experiments with it (see supplementary figure 7). In our hands over a concentration range from nM to mM SQ22536 did not inhibit Forskolin-induced CRE-luciferase activity. In fact, at higher doses (above 30µM) it enhanced it. We have no explanation of this finding and therefore also toned down our statements about involvement of adenylyl cyclase in the pathway investigated.

Figure 9B: Please, include the blot to visualize the data points.

Since this figure is already busy we add the original image of the blot to the supplements (supplementary figure 8).

Minor remarks

Methods Western Blot: How long are the cells kept in serum-free medium before stimulation?

Regarding the SCNCRE cells overnight (appr. 12 hours). SCN ex vivo was directly extracted at the Zeitgeber time indicated.

Figures 2 and 9A,C. Please use lines as in 9B to illustrate significances.

Figure 2: control 24 well

Figure 2 and 9 Lines to illustrate significances and the missing “l” has been added.

Control has been changed to “Co”

Figure 3B/D: The use of similar symbols is very unfavorable. Pleas use different sybols (open and closed and/or colours (see Figure 8). Same holds for 4A,B control and MDLI symbols; 5A;B control and Rp; 6A,B control and IBMX; 7A,B control and probenecid as well as 9D with control and Fsk.

Figure 3A,C: labeling of Y-axis differs from all other figures.

The Y-axis is always the raw RLU readout. In figure 3 A and C the values are rather low but within the range over different cell and luciferin batches. We only used cells between passage 3 and 10 and had to thaw new cells regularly and have no control over the quality of luciferin batches.

Figure 3C: Forskolin concentration in the insert.

Figure 3 C shows the effect of different doses of forskolin. We can write forskolin on the X-axis and delete the insert if wanted.

Figure 3D: what is the meaning of the “k” in the inserted figure legend ?

“k” stands for “kilo=1000”, 25k is 25.000 cells. We added the full data on the issue of the influence of cell number (zero/no cells to 100.000 cells per well) and the CRE-luciferase response to the supplements.

Figure 3 legend: B) Influence of cell number seeded on forskolin-treated and untreated (control) cells. C) Dose-dependent influence of forskolin …. (please ask a native speaker!)

We think we addressed all issues with Figure 3, hope the legend is now written in understandable English and apologize for the poor former wording.

Figure 4 and 5 legend: … by 1 µM (A) and 0.1 mM (B) …… Next sentence “Rp-8-Br-cAMPs displayed…” is done twice.

The superfluous sentence has been deleted.

Figure 9, row 4: -- values of a SCNCRE ….

The “of” after “a” was deleted.

References

  1. O'Neill, J.S.; Maywood, E.S.; Chesham, J.E.; Takahashi, J.S.; Hastings, M.H. cAMP-Dependent Signaling as a Core Component of the Mammalian Circadian Pacemaker. Science 2008, 320, 949–953, doi:10.1126/science.1152506.
  2. Rawashdeh, O.; Jilg, A.; Maronde, E.; Fahrenkrug, J.; Stehle, J.H. Period1 gates the circadian modulation of memory-relevant signaling in mouse hippocampus by regulating the nuclear shuttling of the CREB kinase pP90RSK. J. Neurochem. 2016, 138, 731–745, doi:10.1111/jnc.13689.

Round 2

Reviewer 1 Report

The authors have provided more cAMP measurements to answer the questions about intra and extracellular cAMP conc. dynamics. I think the paper can be accepted now.